# Novel Insights into PARK7 (DJ-1), a Potential Anti-Cancer Therapeutic Target, and Implications for Cancer Progression

**DOI:** 10.3390/jcm9051256

**Published:** 2020-04-26

**Authors:** Wook Jin

**Affiliations:** Laboratory of Molecular Disease and Cell Regulation, Department of Biochemistry, School of Medicine, Gachon University, Incheon 406-840, Korea; jinwo@gachon.ac.kr

**Keywords:** PARK7 (DJ-1), therapeutic target, apoptosis, redox sensor, chemoresistance

## Abstract

The expression of PARK7 is upregulated in various types of cancer, suggesting its potential role as a critical regulator of the pathogenesis of cancer and in the treatment of cancer and neurodegenerative diseases, including Parkinson’s disease, Alzheimer’s disease, and Huntington disease. PARK7 activates various intracellular signaling pathways that have been implicated in the induction of tumor progression, which subsequently enhances tumor initiation, continued proliferation, metastasis, recurrence, and resistance to chemotherapy. Additionally, secreted PARK7 has been identified as a high-risk factor for the pathogenesis and survival of various cancers. This review summarizes the current understanding of the correlation between the expression of PARK7 and tumor progression.

## 1. Introduction

The human PARK7 protein, also known as DJ-1, is a small ubiquitously expressed protein (20 kDa) comprising 189 amino acids. It is a member of the large PARK7/PfpI superfamily [1] and is overexpressed in various tissues, including the testes, sperm, and the brain [2,3,4]. Homozygous deletions and missense mutations in PARK7 lead to the early onset of autosomal recessive Parkinson’s disease (PD) with various symptoms, including dyskinesia, rigidity, and tremors, that account for slow-onset (1–2%) and early-onset PD [5,6,7]. Several mutations are involved in the pathogenesis of PD, which are being increasingly identified, and include the L166P, L10P, T19L, R26A, D149, G78G, M26I, R98Q, D149A, ΔP158, L172Q, and L172G mutations. These mutations reduce the stability of PARK7 and induce loss of function. The DJ-1 mutants, R98Q, D149A, and L166P, are involved in the pathogenesis of PD through induction of mitochondrial dysfunction [8,9,10,11,12]. Additionally, the mutations L166P, L172Q, L10P, and ΔP158 in PARK7 markedly reduce the expression of PARK7 by inducing its rapid degradation [13,14]. The mutations D149A and L166P lead to the loss of p53-mediated cell death, and unlike the L166P mutation, the D149A mutation stabilizes PARK7 by inhibiting cleavage by caspase-6. However, the L166P mutation leads to the loss of function of PARK7 and reduces the levels of PARK7 by inducing rapid proteolysis [15,16,17].

The pathogenicity of PD is also associated with the accumulation of PARK7 aggregates, mediated by the formation of amyloid-β plaques that cause alterations in PARK7 folding, and this aggregation is mediated by the oxidation of the catalytic C106 residue of PARK7 [18,19,20]. In PD, PARK7 forms aggregates with α-synuclein via protein–protein interactions and hydrophobic interactions between PARK7 and α-synuclein [19,21]. The filamentous aggregation of PARK7 by inorganic phosphate and reactive oxygen species (ROS) induces cell death, and this is significantly reduced by treatment with tartrate [22].

Deletion mutations and missense mutations of PARK7 cause loss of function by inducing the aggregation of PARK7. The inhibition of dimer formation by the L166P and R28A mutations does not inhibit the formation of α-synuclein aggregates, owing to the fact that the aggregation of PARK7 causes loss of the molecular chaperone function of PARK7 [16,23,24]. In addition, the mutations L10P, ΔP158, and L166P of PARK7 induce cell death by forming aggregates that impair dimerization [14]. Apart from the implication of PARK7 in the pathogenesis of PD, recent studies on PARK7 emphasize its key importance as an oncogene in the pathogenesis of cancer. 

## 2. Role of PARK7 in Cancer Progression

Apart from its role in neurodegenerative disorders, numerous studies have suggested that PARK7 acts as a mitogen-dependent oncogene that plays a crucial role in the progression of various types of cancer and has been identified as a novel oncoprotein involved in the Ras transduction pathway [25] (Table 1).

The expression of PARK7 is associated with brain tumors. PARK7 is highly expressed in 92.8% of patients with astrocytoma, and this is directly correlated with the aggressiveness of the disease and the poor survival of patients with astrocytoma [26]. Additionally, the expression of PARK7 is upregulated in 85% of patients with glioblastoma [28] and 48.5% of patients with medulloblastoma [28]. The expression of PARK7 is markedly correlated with the increased expression of p-protein kinase B (AKT) and Ki67 and reduces the survival of patients with medulloblastoma.

Apart from being involved in brain tumors, the involvement of PARK7 in other types of human cancers has also been identified. PARK7 has been detected in the cytoplasm of invasive breast cancer cells and is highly expressed in 79% of patients with breast cancer [29]. Functional proteomic profiling identified PARK7 as a novel molecular target of non-small cell lung carcinoma (NSCLC). PARK7 is upregulated in 86% of patients with NSCLC [30], of which 72.2% of patients with NSCLC primarily express PARK7 in the cytoplasm [31]. Another study demonstrated that PARK7 is highly expressed in patients with lung cancer, and its elevated expression is associated with the poor survival and relapse of lung cancer. The nuclear expression of PARK7 stabilizes nuclear factor erythroid 2-related factor (Nrf2) in lung cancer tissues and is significantly associated with the cytoplasmic stability of Nrf2 [29,41].

Several studies have reported the association between the expression of PARK7 and thyroid cancer. The expression of PARK7 is significantly increased in thyroid cancer cells and patients with thyroid cancer. PARK7 is significantly upregulated in 94.6% of patients with thyroid cancer. Additionally, PARK7 is expressed in 100%, 89.5%, 92.3%, and 88.9% of patients with papillary, follicular, medullary, and anaplastic thyroid cancer, respectively [32]. PARK7 is highly expressed in patients with malignant thyroid cancer, indicating that PARK7 may be involved in thyroid cancer [42]. Furthermore, PARK7 acts as a regulatory subunit of an RNA binding protein and is upregulated in 86% of patients with prostate cancer and closely correlated with reduced survival of prostate cancer patients [43] and in prostate cancer cells [43].

The proteins that are related to the process of oncogenesis, including PARK7, oncoprotein 18/Stathmin, and tumor-rejection antigen, are overexpressed in the primary cells of malignant uveal melanoma (UM), which is is the most common primary intraocular malignancy in adults. DJ-1 is also detected in the sera of patients with UM and is secreted by the tumor cells into the bloodstream [44]. It has been reported that the levels of PARK7 are significantly increased in the sera of patients with metastatic UM [45]. Moreover, it has been reported that PARK7 is upregulated in patients with pancreatic carcinoma, and its expression is associated with better differentiation and shorter overall survival [46,47,48]. Another study reported that PARK7 is highly expressed in 52.5% of pancreatic neuroendocrine tumors, and its elevated expression is correlated with aggressiveness, disease progression, and reduced survival of patients with pancreatic neuroendocrine tumors [34].

PARK7 has been identified as a critical regulator of tumor progression in hepatocellular carcinoma (HCC). The expression of PARK7 is elevated in 69.6% of patients with HCC [35], and also in stage III disease rather than the stage I disease. This upregulation is significantly correlated with the poor survival of patients with HCC in subgroups of α-fetoprotein (AFP) <200 ng/mL or tumor size >5 cm. Previous studies have demonstration that PARK7 knockout (KO) mice showed a drastic reduction in both the number of tumors and tumor size by increasing the expression of phosphatase and tensin homolog deleted on chromosome 10 (PTEN) and decreasing the activation of AKT [49,50]. Moreover, PARK7 is highly expressed in hepatitis C virus-infected HCCs [36].

PARK7 is upregulated in 81% of primary ovarian tumors and 80% of solid metastases. The elevated expression of PARK7 is significantly associated with the stage, grade, and poor progression-free survival of ovarian tumors [37], and is highly expressed along with phospho-AKT and mTOR in patients with ovarian cancer and peritoneal malignant effusions [51].

PARK7 is highly expressed in patients with acute leukemia (AL) and leukemia cell lines, compared to that of healthy patients and cells of healthy donors, respectively [52]. Additionally, PARK7 induces the progression of multiple myeloma (MM). PARK7 is markedly increased in patients diagnosed with MM and relapsed patients. The high expression of PARK7 is significantly correlated with the reduced survival of patients with MM [53]. The expression of PARK7 significantly increases in both primary tumors of esophageal squamous cell carcinoma (ESCC) and lymph node metastases, compared to that of the non-neoplastic esophageal epithelium, and there are no differences in the expression of PARK7 between primary tumors of ESCC and lymph node metastases. The increased nuclear translocation of PARK7 enhances metastasis and poor survival of patients with ESCC by activating the PI3K pathway. The nuclear expression of PARK7 is increased in 46% of primary tumors of ESCCs with high distant metastatic potential, compared to primary tumors of ESCCs with low distant metastatic potential. The high levels of nuclear PARK7 in ESCC induces distant metastasis within 1 year of esophagectomy (odds ratio, 3.469) [38].

The levels of PARK7 are significantly increased in patients with clear cell renal cell carcinoma [54] and cervical cancer [55,56]. PARK7 is upregulated in 83.3% of patients with cholangiocarcinoma [39] and in 85% of patients with laryngeal squamous cell cancer [40], which is significantly correlated with the poor survival and tumor recurrence of patients with laryngeal squamous cell cancer [40,57,58]. The loss of PARK7 expression in laryngeal cancer cells reduces their ability to form tumors in vivo and in vitro by increasing the expression of PTEN and suppressing the activation of AKT [59].

The expression of PARK7 is also upregulated in patients with endometrial cancer (EC), and its expression is significantly associated with the pathological progression of EC, including lymph node metastasis, invasion, and differentiation [60,61,62]. Proteomic analysis revealed that the levels of PARK7 are increased in several types of cancer, and O-acetylglucosamine (GlcNAc)-modified PARK7 is involved in the pathogenesis of colorectal cancer (CRC) and human scirrhous-type gastric carcinoma (GC) [63,64,65,66]. It has been reported that the high expression of PARK7 leads to the reduced survival of patients with CRC [67,68] and GC [69].

## 3. Correlation between PARK7 Secretion and the Progression of Carcinoma

Recent studies have demonstrated that PARK7 is secreted into the bloodstream and is involved in the progression of cancer. Highly metastatic breast cancer cells show increased expression of PARK7 and low expression of retained PARK7. In breast cancer, the increased secretion of PARK7 and low expression of retained PARK7 have been more frequently detected in invasive ductal carcinomas than in ductal carcinomas in situ, and this expression pattern is found to be significantly associated with the high histological grade, large tumor size, and reduced survival of patients with breast cancer [70,71,72]. In addition, secreted PARK7 is highly expressed in the serum and nipple fluid of patients with breast cancer and is involved in the regulation of RNA–protein interactions. An increase in the expression of secretory PARK7 in invasive ductal carcinomas may explain its involvement in the pathogenesis of this type of breast cancer.

Secreted PARK7 may be involved in the pathogenesis of other types of cancer. The serum levels of PARK7 are significantly increased in NSCLC, and this increase in the levels of PARK7 has been identified as a high-risk factor that determines the tumor stage and metastasis of NSCLC [73,74]. PARK7 is highly induced, by 2.75 fold, in patients with invasive extrahepatic cholangiocarcinoma (EHCC). Interestingly, the serum levels of PARK7 are higher in EHCC patients with metastasis than in EHCC patients without metastasis [75]. The serum levels of PARK7 are upregulated by nearly 3.26-fold in EC patients and are markedly increased by 2.42-fold in high-risk EC patients compared to that in low-risk EC patients. High serum levels of PARK7 are significantly correlated with the recurrence or poor survival of patients with EC [76]. Moreover, the serum levels of PARK7 are highly elevated in patients with GC [77]. The serum levels of PARK7 serve as a potential diagnostic and prognostic biomarker in pancreatic cancer. A decrease in the serum levels of PARK7 increases gemcitabine-induced apoptosis. The serum levels of PARK7 are significantly increased in patients with pancreatic cancer compared to those in healthy tissues, and this elevation is associated with tumor differentiation and reduced survival of patients with pancreatic cancer [78].

## 4. Functional Roles of PARK7 in Cancer Progression

### 4.1. PARK7 is a Positive Regulator of the Androgen Receptor (AR)-Signaling Pathway

PARK7 significantly increases the transforming activity of cancer cells together with H-Ras/Myc and has been found to be translocated from the cytoplasm to the nucleus in the S phase of the cell cycle [79,80]. The latter finding suggests that the regulation of PARK7 could be essential for the therapeutic intervention of prostatic cancer. The expression of ARs is significantly upregulated by nearly 70-fold in androgen-independent and metastatic prostate cancers [81]. The T877S and H874Y mutations in ARs are induced by treatment with androgen receptor (AR) antagonists such as hydroxyflutamide, and a single mutation in the AR transactivates the AR-signaling pathway in the prostate cancer [82]. PARK7 induces the transcriptional activity of the ARs and is used as a biomarker of prostate cancer. It inhibits the formation of the PIAS2/AR complex by interacting with the AR-binding region of PIAS2 [83]. PARK7 rescues the repression of the transactivation activity of ARs, induced by the EFCAB6/HDAC complex, by forming the PARK7/EFCAB6 complex [84].

### 4.2. PARK7 Suppresses Apoptosis in Cancer Cells

PARK7 expression suppresses the apoptosis of cancer cells (Figure 1A). The incidence of cellular stress by treatment with H_2_O_2_ and mitomycin C (MMC) induces the expression of PARK7, and the apoptosis induced by H_2_O_2_ and mitomycin C is markedly reduced following the increase in PARK7 expression. The overexpression of PARK7 protects prostate cancer cells from apoptosis via the inhibition of tumor necrosis factor-related apoptosis-inducing ligand (TNFSF10) that acts as a specific inducer of apoptosis [43]. PARK7 induces the expression of survivin, which inhibits the apoptotic proteins and induces the proliferation of laryngeal carcinoma cells [85]. PARK7 inhibits autophagy by increasing the formation of the PARK7/MEP3K1 complex that suppresses JNK activity and the transcription of Beclin 1 [86].

PTEN is a multifunctional tumor suppressor protein that suppresses the survival of cancer cells by inhibiting the phosphatidylinositol 3-kinase (PI3K)/AKT signaling pathway [87]. PARK7 induces cancer progression by inhibiting the expression of PTEN. The overexpression of PARK7 significantly reduces apoptosis via the inhibition of various apoptotic stimuli, including UV irradiation, sorbitol, and tumor necrosis factor-α. It has been demonstrated that PARK7 rescues PTEN-induced apoptosis by inducing the activation of the PI3K/AKT signaling pathway, phosphorylation of GSK3β, and expression of cyclin D1 in NIH3T3 cells [29]. PARK7 induces the expression of CTNNB1 by activating AKT and suppressing the expression of PTEN [88]. In addition, PARK7 KO reduces the proliferation of HCCs by inducing the expression of PTEN [50] and inhibition of Interleukin (IL)-6/Signal transducer and activator of transcription 3 (STAT3), mitogen-activated protein kinase (MAPK), and AKT [36]. PARK7 KO in leukemia cells reduces proliferation and induces apoptosis via regulation of the cell cycle and apoptosis-related proteins, including Cdk2, cyclin D1, c-Myc, NF-κB, Bcl-2, and PTEN [52]. PARK7 and the C106S mutant of PARK7 directly interact with PTEN and inhibit the phosphatase activity of PTEN under oxidative conditions. The inhibition of the phosphatase activity of PTEN, the activation of AKT, and the increase in the transforming activity of cancer cells are higher in the presence of the C106S mutant than in the presence of the wild-type PARK7 protein. PARK7 requires the presence of the reduced form of the C106S mutant for inhibiting PTEN activity, inducing proliferation, and increasing the transformation of cancer cells [89]. Lovastatin, which possesses cholesterol-lowering and potential antineoplastic activities, induces apoptosis in ovarian cancer cells by inducing PTEN and reducing phosphor-AKT via inhibition of PARK7 expression [90].

### 4.3. PARK7 Modulates the Expression of Oncoproteins and Tumor Suppressors

PARK7 induces tumor progression by regulating the expression of oncoproteins and tumor suppressor proteins (Figure 1A). DJ-1 induces the SV40 LT-induced immortalization of cells with anchorage-independent growth activity by forming the PARK7/SV40 complex that induces the expression of c-Myc and represses the tumor suppressor activity of TP53 by forming the PARK7/TP53 complex [91]. The expression of PARK7 is upregulated in 85% of glioblastoma tissues and is strongly associated with the nuclear expression of TP53, which in turn is associated with mutations in TP53 and other abnormal perturbations of the TP53 pathway. The expression of PARK7 is negatively associated with the expression of the epithermal growth factor receptor, which correlates with the poor differentiation of glioblastoma cells [27]. PARK7 induces the expression of Telomerase reverse transcriptase (TERT) and telomerase at both the transcriptional and posttranscriptional levels by regulating the PI3K/Akt pathway. Furthermore, PARK7 induces the expression of TERT by increasing the expression of c-Myc [54].

The upregulation of PARK7 enhances tumor formation and the metastasis of pancreatic cancer and disrupts the extracellular matrix of the cancer cells by inducing the expression of the urokinase plasminogen activator by activating Src and ERK [33]. PARK7 also enhances the metastatic ability of breast cancer and MM by inducing epithelial–mesenchymal transition via suppression of the expression of Krűppel-like transcription factor (KLF) 17 and KLF6 [53,92,93]. The suppression of KLF17 by PARK7 induces ID1, which is a negative regulator of PTEN [92].

PARK7 enhances the proliferation and metastasis of CRC via activation of the hedgehog and Wnt signaling pathways. PARK7 increases the proteins involved in the hedgehog signaling pathway, including GLI1, GLI2, and PTCH1. The upregulation of Pleomorphic adenoma gene like-2 (PLAGL2) by PARK7 markedly increases the expression of BMP-4 by inducing the accumulation of nuclear β-catenin and the transcription of T cell factor (TCF), which enhances the activation of the Wnt signaling pathway [68].

### 4.4. PARK7 is an Oxidative-Stress Sensor

PARK7 is a redox sensor that protects cancer cells from oxidative stress. PARK7 induces the expression of the detoxifying enzyme, NAD(P)H quinone oxidoreductase 1 (NQO1), by increasing the stability of the master antioxidant transcription factor, Nrf2. PARK7 suppresses the degradation of GA binding protein transcription factor subunit α (GABPA) by the Kelch-like-ECH-associated-protein 1(KEAP1)-Cullin 3 (CUL3) ubiquitin E3 ligase complex by inhibiting the formation of the GABPA-Keap1 complex and stabilizing GABPA [94].

The expression of PARK7 in thyroid cancer cells leads to the inhibition of apoptosis via suppression of the TNFSF10-induced generation of intracellular ROS [32]. Xanthoangelol, a principal chalcone constituent isolated from the stem exudates of *Angelica keiskei*, induces the apoptosis of neuroblastoma cells by inducing the generation of ROS and reduction of proteins related to oxidative stress, including PARK7, peroxiredoxin 6, triosephosphate isomerase 1, glyceraldehyde-3-phosphate dehydrogenase, and phosphoglycerate mutase 1, which consequently increases oxidative stress and induces apoptosis. It was found that xanthoangelol specifically reduces the expression of PARK7 by 37% compared to that of the control setup. PARK7 protects neuroblastoma cells from apoptosis induced by xanthoangelol-induced oxidative stress [95].

PARK7 has been identified as one of the target proteins of thioredoxin (TXN) in the cardiac tissues of a transgenic murine model overexpressing the cardio-specific lysine methyltransferase 2A (KMT2A) protein. The loss of function of PARK7 occurs by a pathogenic mechanism involving the oxidative damage of PARK7. C53 and C106 residues of PARK7 are primarily oxidized to cysteine sulfonic acid that scavenges the excessive ROS in age-related diseases [96]. However, the oxidation of C53 and C106 in the Tg- KMT2A murine model was reversed by the increase in the expression of KMT2A, via direct interactions between the amino acids and KMT2A. The amino acids are targets of KMT2A and are reduced by the activity of KMT2A [97] (Figure 1B).

### 4.5. PARK7 Acts as an Orchestrator of Cellular Activity by Interacting with Various Proteins

PARK7 enhances the proliferation and aggressiveness of cancer cells by regulating the activity of various proteins via a direct interaction (Figure 2). A study reported that ARs are the interacting partners of PARK7 in prostate cancer cells, and hormonal treatment enhances the formation of PARK7/AR complexes. PARK7 is drastically translocated into the nucleus following treatment with both androgens, including dihydrotestosterone (DHT), and antiandrogens including OH-flutamide and bicalutamide. PARK7 significantly increases the growth of prostate cancer in patients receiving androgen deprivation therapy as neoadjuvant hormone therapy in >6 months [98].

PARK7 interacts with BCL2L1 by increasing its mitochondrial localization in NSCLC cells, which protects against oxidative agents such as ultraviolet B irradiation. The subsequent increase in the stability of mitochondrial BCL2L1 protects against ultraviolet B irradiation-induced cell death. The C106 residue of PARK7 is required for the binding of PARK7 to the BH1-3 domain of BCL2L1 [99]. By microarray using NSCLC cells, PARK7 modulates the nuclear localization of NF-κB by directly interacting with cezanne and inhibiting its activity. Cezanne is a deubiquitination enzyme and a negative regulator of the transcription factor NF-κB that is associated with cell survival. The inhibition of cezanne enhances tumor survival by increasing the expression of IL-8 and ICAM-1 [100].

PARK7 binds to human epidermal growth factor receptor 3 (ERBB3), a therapeutic anti-cancer target, and increases the stability of ERBB3 by inhibiting its ubiquitination. However, the activation of ERBB3 mediated by NRG1 reduces the interaction of ERBB3 with PARK7 in breast cancer cells [101]. Although the overexpression of NRG1 and ERBB3 drastically increases the serum levels of PARK7 and in vivo tumor formation, it markedly reduces the intratumoral expression of PARK7. Moreover, the loss of function of ERBB3 reduces the serum levels of PARK7 in breast cancer cells [102]. Clinical trials have demonstrated that the migration and in vitro tumor formation of cancer cells that overexpress PARK7 are significantly reduced following treatment with anti-ERBB3 antibodies [101].

PARK7 inhibits the activation of MAP3K5 by forming the PARK7/MAP3K5 complex under conditions of mild oxidative stress in order to protect against chemotherapy-induced oxidative stress that leads to apoptosis and autophagy in various cancer cells, including human osteosarcoma, colon cancer, hepatocellular carcinoma, and oral epidermoid carcinoma cells. However, PARK7 is oxidized by excessive levels of ROS and is subsequently dissociated from MAP3K5. The activated MAP3K5 subsequently induces apoptosis by activating the MAP3K5-mediated p38 signaling pathway [103].

### 4.6. PARK7 Acts as a Marker of Chemoresistance

The expression of PARK7 is necessary for the acquisition of chemoresistance in cancer cells. Flutamide treatment stabilizes PARK7, which in turn upregulates the expression of PARK7 [104]. It has been identified that PARK7 is related to cisplatin resistance in NSCLC, small cell lung cancer (SCLC), and clear cell renal cell carcinoma. PARK7 is highly expressed, by more 5.5 fold, in cisplatin-resistant NSCLC cells, and PARK7 KO increases cisplatin sensitivity. PARK7 is highly expressed in patients with NSCLC and has been found to be significantly correlated with the cisplatin resistance and poor survival of patients with NSCLC [105]. PARK7 is associated with the poor survival of patients with SCLC, and markedly decreases the apoptosis of SCLC cells even after treatment with chemotherapeutic drugs, including adriamycin, cisplatin, and etoposide [106]. Proteomic analyses and bioinformatics identified PARK7 as a cisplatin-regulated gene. Cisplatin treatment reduces the expression of PARK7; however, PARK7 inhibits the cisplatin-induced apoptosis of renal cell carcinoma cells [107].

The upregulation of PARK7 expression induces chemoresistance to epirubicin, an anthracycline chemotherapeutic drug for GC, by modulating the epirubicin-induced autophagy [108]. PARK7 is highly expressed in vincristine-resistant GC cells. Vincristine resistance, combined with the overexpression of PARK7 in GC cells, increases cell survival, aggressiveness, and chemoresistance by inducing the expression of P-glycoprotein and BCL2, even during treatment with vincristine, adriamycin, 5-fluorouracil, and cisplatin by increasing the efficiency of pumping chemotherapeutic drugs out of the cell-matrix and inhibiting the apoptosis of GC cells [109].

The expression of PARK7 is significantly correlated with high rates of complete pathological remission following chemotherapy with trastuzumab in patients with breast cancer. Low expression of PARK7 has been detected in 79.6% of pathological complete remission cases, which is defined by the absence of in situ and axillary node invasion following trastuzumab treatment [71]. The loss of function of PARK7 increases adriamycin sensitivity and reduces the activation of AKT in breast cancer cells [110].

Dihydroartemisinin (DHA) exhibits anti-cancer activities, and its anti-cancer activities are potentiated when combined with cisplatin, via the induction of apoptosis and autophagy in cisplatin-resistant ovarian cancer cells [111]. DHA-resistant cell lines show resistance to chemotherapeutic drugs including taxol, etoposide, topotecan (TPT), and 5-flurouracil (5-FU). Proteomic studies and pathway analysis revealed that PARK7 is highly expressed in DHA-resistant cell lines, and the upregulation of PARK7 protects cancer cells from DHA-induced ROS and apoptosis by increasing the stability of Nrf2. This transcription factor induces the expression of several antioxidative proteins, including GCLC and catalase [112]. PARK7 is upregulated in bortezomib-resistant MM cells by upregulating the expression of c-Myc, Hypoxia Inducible Factor 1 Subunit α (HIF1A), Vascular endothelial growth factor (VEGF), and Glucose transporter 1 (GLUT1) [50].

## 5. Modulators of PARK7 Expression in Cancer Progression

### 5.1. Positive Modulators of PARK7 Expression

Recent studies have identified positive modulators of PARK7 expression that enhance tumor progression. Abstrakt has been identified as a PARK7 binding protein by employing a yeast two-hybrid screening assay. Abstrakt is an RNA helicase that is broadly expressed in all tissues. Abstrakt colocalizes with PARK7 in the nucleus and thereby enhances the transforming activity of PARK7 [113]. HSPA5, a promoter of metastasis, induces the expression of PARK7 and peroxiredoxin 4 (PRDX4). The increase in the expression of PARK7 induces cell proliferation in lung adenocarcinoma cells, which have a high invasive ability [114]. The expression of STAT5A in leukemic pre-B cells increases the expression of PARK7 expression, which prevents spontaneous apoptosis and Fas-induced apoptosis [115].

Several proteins can positively regulate the expression of PARK7 by increasing its stability. The increase in the stability of PARK7 by striatin 3 (STRN3) enhances the survival of cancer cells. STRN3 interacts with AKT and PARK7 via the WD-40 domains and a region further upstream of the WD-40 repeats, respectively [116]. STRN3 also increases the expression of PARK7 by inhibiting the proteasome-mediated degradation of PARK7 and subsequently activates AKT by enhancing the stability of PARK7 and increasing the formation of protein-protein complexes for protecting cancer cells from oxidative stress [117]. Moreover, the stability of PARK7 is markedly increased by the loss of function of TP53, and PARK7 induces the activation of TP53 during oxidative stress. The stability of PARK7 is reduced in a negative feedback loop by the upregulation of TP53, which acts as a negative regulator of PARK7 via a post-transcriptional route and eventually inhibits the transformation and proliferation of cells by suppressing the activation of AKT [118].

### 5.2. Negative Modulators of PARK7 Expression

Several negative regulators inhibit the expression of PARK7 for suppressing tumor progression. Treatment with the DNA-damaging agent, MMC, induces TP53 in HCT116 TP53+/+ cells, and the upregulation of TP53 induces the phosphorylation of PARK7 by MAPK8 or via a downstream target of a MAPK8-mediated, transcription-independent event, indicating that the phosphorylation of PARK7 mediated by TP53 may inhibit its function, and subsequently facilitate the apoptosis of CRC cells [119]. PARK7 is a substrate of matrix metalloproteinase (MMP)14, such as LGALS1, which is a potential therapeutic anti-cancer target and is cleaved at more than two sites by MMP1, MMP2, MMP8, MMP9, and MMP14. The cleavage of PARK7 by MMPs may play a role in the pathogenesis of cancer [120]. In contrast, the PARK7-mediated activation of AKT promotes the aggressiveness of GC cells by increasing the expression of MMP2 and MMP9 [121]. Moreover, HSPA5 induces the apoptosis of pancreatic cancer cells by activating the eukaryotic translation initiation factor-alpha/C/EBP homologous protein (eLF2a/CHOP) pathway and reducing the expression of DK-1, prohibitin, CUL3, and ANXA2 [122]. A study revealed that the treatment of PC-3M cells with prostate-specific antigen, a serine protease with anti-angiogenic activity, reduced the expression of oncoproteins, including PARK7, the laminin receptor, vimentin, and Hsp60 [123].

## 6. Role of microRNAs (miRNAs) in the Regulation of PARK7 Expression

MiRNAs are a type of non-coding RNA that is involved in the regulation of PARK7 expression. A study demonstrated that the expression of miR-128-3p is markedly reduced in HCC cells, whereas the expression of PARK7 is upregulated. The miR-128-3p miRNA directly reduces the expression of PARK7 in HCC cells by binding to the 3′-untranslated region of PARK7. The regulation of PARK7 by miR-128-3p induces the sorafenib sensitivity of HCC cells that is mediated via the suppression of the PI3K/AKT pathway [124]. The expression of miR-142 is also significantly reduced in patients with pancreatic cancer, and this inhibits the expression of HIF1A and PARK7, which enhances the sensitivity of the pancreatic cancer cells to adriamycin [125,126]. Moreover, the expression of miR-203 is markedly reduced in pancreatic cancer cells, and the reduction is more significant in cisplatin-resistant pancreatic cancer cells. In contrast, it has been identified that the expression of PARK7 is opposite to that of miR-203. The miR-203 miRNA reduces the proliferation and cisplatin resistance of pancreatic cancer cells by inducing the expression of PTEN and suppressing the activation of PARK7 and AKT [127].

## 7. Therapies Targeting PARK7

Recently, several candidates that target the activity or expression of PARK7 have been investigated. Diallyl disulfide (DADS) has anti-cancer potential, and treatment with DADS induces apoptosis and inhibits the metastatic potential of leukemia cells by suppressing the expression of oncoproteins, including PARK7 [128]. DADS inhibits the activation of Src and focal adhesion kinase (FAK) by reducing the expression of PARK7 [129].

Dimethyl fumarate, a derivative of fumaric acid that is used to treat patients with relapsing multiple sclerosis and psoriasis, has been recently used in clinical trials for the treatment of glioblastoma, lymphoma, and leukemia [130]. Treatment with dimethyl fumarate suppresses the tumor formation and proliferation of CRC cells by inducing oxidative stress and reducing the intracellular levels of glutathione, by reducing the expression of PARK7, which depletes GABPA [131]. Additionally, dimethyl fumarate induces the apoptosis of cancer cells by inhibiting the induction of GABPA/PARK7 by mutated Kirsten rat sarcoma 2 viral oncogene homolog (KRAS) [132]. 

Treatment with paclitaxel inhibits the proliferation and metastatic potential of breast cancer cells by inducing the expression of KLF17 and suppressing the expression of snail, PARK7, and ID1, which results in the inhibition of epithelial-mesenchymal transition [133]. PARK7 cooperates with Ras to enhance the transformation of NIH3T3 cells and inhibits apoptosis following treatment with both paclitaxel and a MEK inhibitor [30].

Ciclopirox olamine (CPX) has antitumor activity and has recently been identified as an anti-cancer candidate. Ciclopirox olamine inhibits the growth of CRC cells by inducing mitochondrial dysfunction and the accumulation of ROS by reducing the expression of PARK7 by Runt-related transcription factor 1 (RUNX1) [134].

Adenovirus (Ad)-mediated gene therapies using tumor suppressor genes such as interleukin 24 (IL24) suppresses the growth of NSCLC cells [135]. The expression of IL24 by Ad- IL24 induces the apoptosis of cervical cancer cells by regulating the expression of proteins related to apoptosis, including PARK7 [136].

Potent 1H-indole-2,3-dione derivatives have recently developed as a compound inactivating the critical Cys106 residue of PARK7 by binding potent PARK7 inhibitors to a conspicuous pocket (Cys106) of PARK7. The maximal affinity and inhibitory potency (IC_50_) of PARK7 inhibitors were 100 and 300 nM, respectively [137]. The continuous studies using these PARK7 inhibitors may need more evaluations for the therapeutic application of cancers and neurodegenerative diseases.

## 8. Conclusions

Consistent with its implication in various pathological processes, including PD, several lines of evidence indicate that the expression of PARK7 is involved in the clinicopathology of various types of cancer. As described in this review, recent studies on the expression of PARK7 have provided fundamental insights into the mechanisms underlying the functions of PARK7 in cancer progression, including the suppression of apoptosis, redox sensing, and serving as a marker of chemoresistance. PARK7 has been identified as a potential therapeutic target for the treatment of neurodegenerative diseases and cancer. Nevertheless, the functions of PARK7 in cancer progression remain mostly unknown, necessitating the characterization of PARK7 in the pathogenesis of cancer by further studies. These future investigations may elucidate and identify the novel functions of PARK7, which will provide researchers with an improved understanding of its functions. The approach of molecular pathologic epidemiology (MPE), which is the epidemiology-based molecular classification of cancer, is essential for a better understanding of the link between PARK7 and cancer progression. The research framework of MPE in cancer provides integration data science of the multidisciplinary cancer field with endogenous cancer risk factors (somatic genetic and epigenetic alterations, tumor-immune systems interactions, tumoral molecular mechanisms, and response to therapeutic drugs), which are influenced by the exogenous cancer risk factors in the pathogenesis of cancer, including diet, nutrition, lifestyle, the environment, microbial exposures (gut and intestinal microbiota), and other exogenous factors. Moreover, computational digital pathology, systems biology, and artificial intelligence can be used to support the framework of MPE in cancer and other diseases [138,139]. Identification of the novel regulatory functions underlying the acquisition of cancer stem cells and drug resistance by PARK7 through the application of the MPE in cancer progression will provide novel and practical information that could aid the development of novel therapeutic strategies.

## Figures and Tables

**Figure 1 jcm-09-01256-f001:**
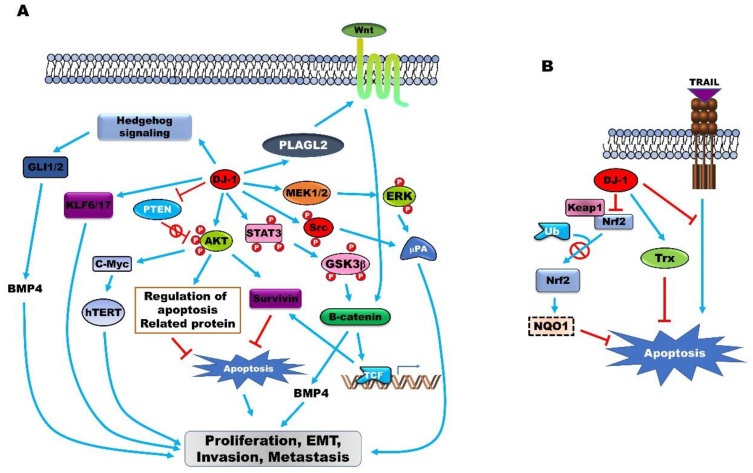
(**A**) PARK7 induces tumor progression by permitting tumor initiation, continued proliferation, metastasis, recurrence, and resistance to chemotherapy by activating several intrinsic cell signaling pathways and upregulating the expression of oncoproteins. PARK7 activates the Hedgehog, Wnt, AKT, extracellular-signal-regulated kinase (ERK), and STAT3 signaling pathways and induces the expression of Krűppel-like transcription factor (KLF6/KLF17), which subsequently enhances tumor progression. (**B**) The upregulation of PARK7 expression serves as a sensor of oxidative stress and protects cancer cells from oxidative stress-induced apoptosis by inducing GABPA and thioredoxin (TXN), or by suppressing the TNFSF10-induced generation of intracellular reactive oxygen species (ROS).

**Figure 2 jcm-09-01256-f002:**
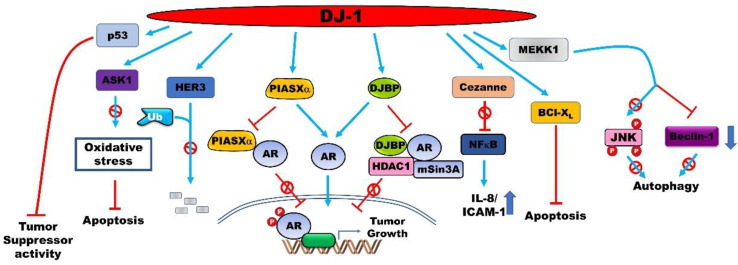
PARK7 enhances the proliferation and aggressiveness of cancer cells by regulating the activity of several proteins via direct interactions. PARK7 binds to oncoproteins and tumor suppressors to regulate their activation, stabilization, and cellular localization. This subsequently enhances the metastatic potential of cancer cells and protects them from apoptosis and autophagy.

**Table 1 jcm-09-01256-t001:** Involvement of PARK7 in various types of cancer.

Type of Cancer	Frequency	Refs
Astrocytoma	103/111 (92.8%)	[26]
Glioma	34/40 (85%)	[27]
Medulloblastomas	32/66 (48.5%)	[28]
Breast cancer	22/28 (79%)	[29]
Non-small cell lung carcinoma	6/7 (86%) 13/18 (72.2%)	[30,31]
Thyroid cancer	70/74 (94.6%)	[32]
Prostate cancer	66/76 (86%)	[33]
Pancreatic neuroendocrine tumors	21/40 (52.5%)	[34]
Hepatocellular carcinoma Hepatitis C virus-infected hepatocellular carcinoma (HCC)	32/46 (69.6%) 30/32 (93.75%)	[35] [36]
Ovarian cancer	63/72 (87%)	[37]
Esophageal squamous cell carcinoma	11/21 (46%)	[38]
Cholangiocarcinoma	5/6 (83.3%)	[39]
Laryngeal squamous cell cancer	51/60 (85%)	[40]

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
