# Peer review of "Novel Insights into PARK7 (DJ-1), a Potential Anti-Cancer Therapeutic Target, and Implications for Cancer Progression"

_jcm, 2020, doi:10.3390/jcm9051256_

Round 1

Reviewer 1 Report

This is a generally well-written review on the roles of PARK7 (DJ-1) in cancer. The topic is of interest. This paper can be improved in following points.

Many discussion points are binary and dichotomous (eg, signal upregulation vs. downregulation; bacteria positive vs. negative; cell growth vs. no growth). It is a bit misleading because many biological characteristics form or represent continuum. 

There are many environmental, dietary, and lifestyle factors that influence the microbiome, immune system, carcinogenic mechanisms, and response to therapy. The authors should discuss these points; influence of those factors, eg, smoking, diet, obesity, etc.

There are also influences of germline and somatic genetic variations on both oncogenic signaling and immune function/microbiota. Those can be discussed. 

In these contexts, as a future direction, research on dietary / lifestyle / genetic factors, microbiome, immunity, and molecular tissue biomarkers is needed. The authors should discuss molecular pathological epidemiology (MPE), which can investigate those factors in relation to molecular pathologies, immunity, and clinical outcomes in cancer. MPE, its strengths and challenges have been discussed in Gut 2011; Annu Rev Pathol 2019, etc. I believe MPE research can be a promising direction.

Please use exact symbols for gene products (all proteins) set by the HUGO committee (www.genenames.org). All proteins come from genes. Many proteins are mentioned in the text and figures. All protein names must be replaced with official symbols (plus colloquial names in parenthesis). Eg, “PARK7 (DJ-1)”, “CTNNB1 (beta-catenin)”, etc.

Colloquial names such as so-called DJ-1 can be used but only in parenthesis. DJ-1 should not be used alone.

Please use “PARK7 (DJ-1)”, PARK7 protein, PARK7 gene, etc. 

Please revise title using the term “PARK7 (DJ-1)”. 

See https://www.genenames.org/data/gene-symbol-report/#!/hgnc_id/HGNC:16369

Approved symbol 

PARK7

Approved name 

Parkinsonism associated deglycase

Locus type 

gene with protein product

HGNC ID 

HGNC:16369

Symbol status 

Approved

Previous names 

 Parkinson disease (autosomal recessive, early onset) 7 

 parkinson protein 7 

Alias symbols 

DJ-1

DJ1

GATD2

Chromosomal location 

1p36.23

Author Response

his is a generally well-written review on the roles of PARK7 (DJ-1) in cancer. The topic is of interest. This paper can be improved in following points.

Many discussion points are binary and dichotomous (eg, signal upregulation vs. downregulation; bacteria positive vs. negative; cell growth vs. no growth). It is a bit misleading because many biological characteristics form or represent continuum. 

There are many environmental, dietary, and lifestyle factors that influence the microbiome, immune system, carcinogenic mechanisms, and response to therapy. The authors should discuss these points; influence of those factors, eg, smoking, diet, obesity, etc.

There are also influences of germline and somatic genetic variations on both oncogenic signaling and immune function/microbiota. Those can be discussed. 

 In these contexts, as a future direction, research on dietary / lifestyle / genetic factors, microbiome, immunity, and molecular tissue biomarkers is needed. The authors should discuss molecular pathological epidemiology (MPE), which can investigate those factors in relation to molecular pathologies, immunity, and clinical outcomes in cancer. MPE, its strengths and challenges have been discussed in Gut 2011; Annu Rev Pathol 2019, etc. I believe MPE research can be a promising direction.

Thank you for your recommendation. As your suggestions, we added the this contents and references in Conclusion as shown below.

The approach of molecular pathologic epidemiology (MPE), which is the epidemiology-based molecular classification of cancer, is essential for a better understanding of the link between the PARK7 and the cancer progression. The research framework of MPE in cancer provides integration data science of the multidisciplinary cancer field with endogenous cancer risk factors (somatic genetic and epigenetic alterations, tumor-immune systems interactions, tumoral molecular mechanisms, and response to therapeutic drugs), which influenced by the exogenous cancer risk factors in the pathogenesis of cancer, including diet, nutrition, lifestyle, the environment, microbial exposures (gut and intestinal microbiota), and other exogenous factors. Moreover, computational digital pathology, systems biology, and artificial intelligence can be used to support the framework of MPE in cancer and other diseases [139,140]. Identification of the novel regulatory functions underlying the acquisition of cancer stem cells and drug resistance by PARK7 through the application of the MPE in cancer progression will provide novel and practical information that could aid the development of novel therapeutical strategies.

Please use exact symbols for gene products (all proteins) set by the HUGO committee (www.genenames.org). All proteins come from genes. Many proteins are mentioned in the text and figures. All protein names must be replaced with official symbols (plus colloquial names in parenthesis). Eg, “PARK7 (DJ-1)”, “CTNNB1 (beta-catenin)”, etc.

 Colloquial names such as so-called DJ-1 can be used but only in parenthesis. DJ-1 should not be used alone.

 Please use “PARK7 (DJ-1)”, PARK7 protein, PARK7 gene, etc. 

 As your suggestions, we checked and corrected this and other proteins using the HUGO committee.

Please revise title using the term “PARK7 (DJ-1)”. 

 As your suggestions, we corrected this and other proteins using the HUGO committee.

See https://www.genenames.org/data/gene-symbol-report/#!/hgnc_id/HGNC:16369

Approved symbol 

PARK7

Approved name 

Parkinsonism associated deglycase

Locus type 

gene with protein product

HGNC ID 

HGNC:16369

Symbol status 

Approved

Previous names 

 Parkinson disease (autosomal recessive, early onset) 7 

 parkinson protein 7 

Alias symbols 

DJ-1

DJ1

GATD2

Chromosomal location 

1p36.23

Reviewer 2 Report

In this manuscript, Jin W investigates the role of DJ-1 expression in tumor progression. The manuscript is well written and is based on a bibliography sufficiently updated. 

 My major concern is about Cap 2., which I found too long and sometimes repetitive. I suggest deleting from Tab1 the frequency percent of the last three types of cancer indicated and removing the corresponding written part at the end of Cap 2. Moreover, why didn’t you report in the Tab1 the percent frequency of Prostatic cancer and Leukemia, since they are among cancer you reported for the most in this review? If possible try to complete the Tab1. 

 Paragraph 4.5: specify which cancer o cellular system you are referring to, for each different pathway mentioned. 

 Cap 7. Consider to include this reference: https://www.ncbi.nlm.nih.gov/pubmed/30063823. The authors present the first DJ-1 inhibitor with very impressive properties (such as affinity for the specific target-DJ-1- and IC50). Even if they propose this molecule for the treatment of Parkinson's disease, you mention this research 

Author Response

In this manuscript, Jin W investigates the role of DJ-1 expression in tumor progression. The manuscript is well written and is based on a bibliography sufficiently updated. 

 My major concern is about Cap 2., which I found too long and sometimes repetitive. I suggest deleting from Tab1 the frequency percent of the last three types of cancer indicated and removing the corresponding written part at the end of Cap 2. Moreover, why didn’t you report in the Tab1 the percent frequency of Prostatic cancer and Leukemia, since they are among cancer you reported for the most in this review? If possible try to complete the Tab1. 

Thank you for your recommendation. As your suggestions, we deleted corresponding written part at the end of Cap 2 and frequency percent of the last three types of cancers (Supraglottic squamous cell carcinoma, Urothelial carcinoma, Nasopharyngeal carcinoma) from Table 1 as shown below.

PARK7 acts as a regulatory subunit of an RNA binding protein and is upregulated in 86% of patients with prostate cancer and closely correlated with reduced survival of prostate cancer patients and in prostate cancer cells.

Also, we added the percent frequency of prostatic cancer. Moreover, we checked the frequency of DJ-1 expression on leukemia patients using the previous study to add the frequency of leukemia as your suggested. However, only one previous paper, which we already referenced, stated that the DJ-1 was upregulated in patients with myelodysplastic syndromes (MDS) and acute myeloid Leukemia (AML), but the frequency of DJ-1 does not mentioned in this previous paper. So, we could not confirm the frequency of DJ-1 in Leukemia.

Paragraph 4.5: specify which cancer o cellular system you are referring to, for each different pathway mentioned. 

As your suggestions, we added cancer or cellular systems in paragraph 4.5, which we are referring to.

Cap 7. Consider to include this reference: https://www.ncbi.nlm.nih.gov/pubmed/30063823. The authors present the first DJ-1 inhibitor with very impressive properties (such as affinity for the specific target-DJ-1- and IC50). Even if they propose this molecule for the treatment of Parkinson's disease, you mention this research 

As your suggestion, we added the paragraphs with DJ-1 inhibitors as shown below.

Potent 1H-indole-2,3-dione derivatives have recently developed as a compound inactivating the critical Cys106 residue of PARK7 by binding potent PARK7 inhibitors to a conspicuous pocket (Cys106) of PARK7. The maximal affinity and inhibitory potency (IC50) of PARK7 inhibitors were 100 and 300 nM, respectively. The continuous studies using these PARK7 inhibitors may need more evaluations for the therapeutic application of cancers and neurodegenerative diseases.